# Dynamic Layer Tying for Parameter-Efficient Transformers

**Tamir David- Hay & Lior Wolf**
Blavatnik School of Computer Science, Tel Aviv University
`{davidhay,wolf}@mail.tau.ac.il`

## Abstract

In the pursuit of reducing the number of trainable parameters in deep transformer networks, we employ Reinforcement Learning to dynamically select layers during training and tie them together. Every few iterations, the RL agent is asked whether to train each layer $i$ independently or to copy the weights of a previous layer $j < i$. This facilitates weight sharing, reduces the number of trainable parameters, and also serves as an effective regularization technique. Experimental evaluations validate that our model modestly outperforms the baseline transformer model with regard to perplexity and drastically reduces the number of trainable parameters. In particular, the memory consumption during training is up to one order of magnitude less than the conventional training method.

## 1 Introduction

The recent work on large language models is based mostly on the transformer architecture of Vaswani et al. (2017). Such models have become increasingly larger and are trained for 100s of thousands of GPU hours using high-end GPUs (Brown et al., 2020; Chowdhery et al., 2022; Rae et al., 2021; Touvron et al., 2023).

However, it is clear that the Transformer architecture (like other deep architectures) is overparameterized. For example, pruning can be used to reduce the number of FLOPs of transformers during inference time at least by half, with little effect on accuracy (Kurtic et al., 2022; Kwon et al., 2022), attention heads can be removed post-training with little effect on performance (Michel et al., 2019; Voita et al., 2019). The lottery ticket hypothesis holds for transformers (Frankle & Carbin, 2018; Chen et al., 2020a;b; Prasanna et al., 2020; Movva & Zhao, 2020), and, perhaps most relevant to our work, layers can be dropped altogether during inference Fan et al. (2019); Sajjad et al. (2020), and attention scores can be reused Bhojanapalli et al. (2021).

Motivated by the potential to reuse transformer layers, we conducted a preliminary experiment in which we started with a transformer of $L$ layers and trained only $\frac{L}{2}$ layers by sharing the weights between layers $i$ and layer $i + \frac{L}{2}$ for $i < \frac{L}{2}$. The transformer trained this way achieved the same, or somewhat better performance, as the conventional $L$ layer transformer.

This encouraging preliminary finding raises a few questions. First, is there something special about this pattern of repetition? Second, is a factor of two the best we can get? Taken to the extreme, it would be desirable that every layer in the architecture either replicates one of the previous layers or, if needed for the sake of accuracy, have a new set of weights.

Our method opts to find such a general pattern. Trying to train only once, we view the repetition pattern $\boldsymbol{a}$ as a dynamic action that some driver network $\mathcal{Q}$ learns from reinforcement during the training of the primary network $\mathcal{T}$. Every few epochs, a new action vector $\boldsymbol{a}$ is obtained based on the Q-function estimation given by $\mathcal{Q}$.

The element $a_i \in [0, 1, \ldots, i]$ for $i = 1, \ldots, L$ indicates from which layer to copy the weights to layer $i$ of $\mathcal{T}$. If $a_i = i$ then the weights of this layer are being optimized independently of other layers. If $a_i < i$ then the weights of layer $a_i$ are used as the weights of layer $i$. This is transductive: if layer $a_i = j$ and $a_j = k$, then both layers $i$ and $j$ share the same weights of layer $k$.

After a few training iterations of the primary network $\mathcal{T}$, the reward for the driver network $\mathcal{Q}$ is computed by considering the loss obtained on a few training batches. $\mathcal{Q}$ is then updated, and a new action vector $\boldsymbol{a}$ is recovered. Depending on the dynamics of the driver network, the changes in the replication pattern can be rather rapid. Yet, as we show, the training process is stable.

Our results indicate that training this way leads to a replication of at least 75% of the transformer layers while maintaining the same level of accuracy, or even slightly better, as the full $L$ layer transformer. This is achieved with a relatively small $\mathcal{Q}$ network, which is only applied during training.

Our contributions are: (i) Presenting a novel method for dramatically reducing the number of parameters in a transformer architecture. (ii) Establishing the potential of Reinforcement Learning (RL) to serve as a pivotal mechanism for dynamically optimizing the architectural configurations of transformers during training. The impact of RL in this context is considerably more profound than its conventional applications, such as adaptive learning rate tuning Xu et al. (2019). (iii) Demonstrating the use of RL in Neural Architecture Search (NAS) in a single training pass, unlike all previous work we are aware of, which follow Baker et al. (2017); Gao et al. (2019); Zoph & Le (2016) and collect multiple training sessions. (iv) Showing that transformers can be trained effectively, despite rapid changes in architecture during the training process.

## 2 RELATED WORK

Our method changes the architecture of the Transformer network and is, therefore, a Neural Architecture Search (NAS) method. The promise of the field is to discover architectures that would surpass human-designed ones in performance. While most recent contributions rely on techniques such as Differentiable Architecture Search (Liu et al., 2018a), some of the earlier approaches relied on RL. Zoph & Le (2017), employ a recurrent neural network (RNN) to generate architectural descriptions of neural networks and train it with RL. Baker et al. (2017) employ Q-learning to search for optimal CNN architectures. Cai et al. (2018) uses a controller, trained with the policy gradient method, to search for architectures in a more computationally efficient manner. As mentioned, RL NAS methods suggest a fixed architecture and train it from scratch, using the validation score as a reward. The trained network is not changed dynamically during training as we do.

The use of RL for dynamically controlling the training of a deep neural network has focused on learning rate optimization. Controlling the learning rate is often done with a fixed schedule, such as a step decay or a cosine decay, which determines the step size for each iteration of the optimization process (Ruder, 2016). Xu et al. (2019) employ proximal policy optimization (PPO) (Schulman et al., 2017) trained across multiple sessions (not a single session as in our method). (Subramanian et al., 2023) also employ PPO, and use a state vector that includes the training loss of the last epoch, the epoch index, and the number of remaining epochs.

Considerable effort has been dedicated to making transformers more efficient by reducing the quadratic complexity of the self-attention mechanism, e.g., (Child et al., 2019; Ma et al., 2021). With respect to parameter efficiency, network pruning methods (Molchanov et al., 2016; Hassibi et al., 1993; Frankle & Carbin, 2018; Liu et al., 2018b) including the transformer pruning methods mentioned above (Kurtic et al., 2022; Kwon et al., 2022) reduce the size of the network by removing or shrinking matrices from the network. Such methods often require further re-training, while our method is applied during training, maintaining the training time per epoch and reducing the peak memory consumption. The recent Wanda method (Sun et al., 2023) performs straightforward magnitude-based pruning (Han et al., 2015; Gale et al., 2019; Zhu & Gupta, 2018; Liu et al., 2018b) on the trained transformer. Despite its simplicity, it is shown to outperform other pruning alternatives. In comparison to our method, the sparsity demonstrated is up to 50% of the weights, while our method is shown to reduce 75% to 87% of the parameters. Our approach, which focuses on reuse, and pruning, which attempts to "reduce", are not mutually exclusive and can be combined.

Other methods that reuse computations or parameters within transformers include the Reuse Transformer (Bhojanapalli et al., 2021) which, unlike our method, uses a specific and fixed pattern of reusing elements and only reuses attention heads. Overall less than 10% of the parameters are shared. Similarly to the Reuse Transformer, the Subformer (Reid et al., 2021) shares the parameters of the middle layers, however, much more extensively, reaching up to 50% reduction in the number of parameters. This requires the addition of auxiliary network elements, which we do not do.

---

**Algorithm 1** Q-learning driven dynamic layer tying

---

**Require:** $L$ the number of layers, $K$ the number of training steps of $\mathcal{T}$, $k$ the number of training steps between the update and evaluation of $\mathcal{Q}$, $\gamma$ the discount factor, and $\epsilon$ initial exploration probability
  1: Initialize the primary model $\mathcal{T}$ and the Q-network $\mathcal{Q}$
  2: Freeze layers 1 to $L-1$ in $\mathcal{T}$, such that only layer 0 trains at initialization.
  3: Initialize $\boldsymbol{s} = \boldsymbol{a} = 0$                                                    ▷ An all zero vector
  4: **for** step = 0 to $K-1$ **do**
  5:     Sample a mini-batch $B$ from the dataset
  6:     Perform a training step with $\mathcal{T}$ on $B$
  7:     **if** mod(step,k) == 0 **then**                                      ▷ Every $k$ steps
  8:         Obtain an action vector $\boldsymbol{a} = \pi(\boldsymbol{s})$
  9:         Compute $\boldsymbol{s}'$ based on $\boldsymbol{a}$                                        ▷ Eq. 1
10:         **for** $i = 0$ to $L-1$ **do**
11:             **if** $\boldsymbol{s}'_i \neq \boldsymbol{s}_i$ **then**
12:                 **if** $\boldsymbol{s}'_i == i$ **then**
13:                     Untie layer $i$ of $\mathcal{T}$ ▷ Copy its weights and update it independently of layer $\boldsymbol{s}_i$
14:                 **else**
15:                     Replicate all weights of layer $\boldsymbol{s}'_i$ of $\mathcal{T}$ to layer $i$ of $\mathcal{T}$
16:                     Tie the weights of layer $i$ to layer $\boldsymbol{s}'_i$
17:                 **end if**
18:             **end if**
19:         **end for**
20:         Sample a mini-batch $B$ from the data-set
21:         $r_{step}$ = Compute negative PPL score based on $\mathcal{T}$ on $B$
22:         $r_{predicated} = \mathcal{Q}(\boldsymbol{s}, \boldsymbol{a})$                                    ▷ Eq. 3
23:         $r = r_{step} + \gamma * \max_a \mathcal{Q}(\boldsymbol{s}')_{\boldsymbol{a}}$
24:         $L = MSE(r_{predicted}, r)$
25:         update $\mathcal{Q}$ using $L$
26:         $\boldsymbol{s} = \boldsymbol{s}'$
27:         $\epsilon = \max\{\epsilon * 0.95, 0.1\}$
28:     **end if**
29: **end for**

---

Takase & Kiyono (2021) explore three different fixed patterns of sharing parameters, reusing 50% to 66% of the layers. The differences in performance between the patterns are small, and our last ablation (ablation vii) is similar to the Cycle pattern. Xiao et al. (2019) share attention weights (and not the parameters for computing these), based on the attention similarity. The number of reduced parameters is not reported but the average speedup is 1.3 (23% reduction).

Parameter Efficient Fine-Tuning (PEFT) often target specific layers or modules, e.g., only the top layers (Gheini et al., 2021), only the bias parameters (Zaken et al., 2021), or selecting based on scores (Sung et al., 2021; Vucetic et al., 2022). Additive PEFT methods introduce additional trainable parameters that can be added to the attention and feed-forward layers of transformers (Houlsby et al., 2019). LoRA (Hu et al., 2022) adds low-rank matrices to the weight matrices. PEFT methods substantially reduce the number of trainable parameters, but are applicable for finetuing (after the full model has been trained), while our method is for training from scratch. See Sec. 5 for future work on finetuning.

## 3   METHOD

We aim to train a transformer $\mathcal{T}$ with $L$ layers from scratch. All elements of a transformer layer, including the key, query, and value projections, and the linear layers are considered as a single set of training parameters. The set of parameters for layer $i$ can be either independent from all layers $j < i$, or tied to the set of parameters of some layer $j < i$.

The state vector $\boldsymbol{s} \in \mathbb{N}^L$ indicates, at each location $i = 0, 1, \ldots, L-1$, the layer with the lowest index that has the same tied weights. Therefore, $\forall i \in [0, \ldots, L-1] : 0 \leq \boldsymbol{s}_i \leq i$. If $\boldsymbol{s}_i = i$

it indicates that layer $i$ does not have its parameters tied with any of the previous layers. By this definition, it always holds that $\boldsymbol{s}_0 = 0$.

The action space is similar, except that the action vector $\boldsymbol{a} \in \mathbb{N}^L$ can point to any previous layer that has its weights tied with layer $i$, not necessarily the one with the lowest index $j \leq i$.

To obtain $\boldsymbol{s}$ from $\boldsymbol{a}$, one can employ the following recursion

$$\boldsymbol{s}_i = \begin{cases} i & \boldsymbol{a}_i == i \\ \boldsymbol{s}_{\boldsymbol{a}_i} & \text{Otherwise} \end{cases} \tag{1}$$

The Q-function of a Markov Decision Process represents the expected cumulative future reward for taking a particular action $\boldsymbol{a}$ a in a particular state $\boldsymbol{s}$, while following a certain policy $\pi$ (Sutton & Barto, 2018). Similarly to previous work that employs deep Q-learning(Mnih et al., 2013), we employ an $\epsilon-$greedy policy obtained interpolating between a random policy and one obtained by maximizing, at a given state, the Q-function over the available actions.

$$\pi(\boldsymbol{s}) = \begin{cases} \arg\max_{\boldsymbol{a}} \mathcal{Q}(\boldsymbol{s}, \boldsymbol{a}) & \text{at probability } 1 - \epsilon \\ \text{a uniformly sampled } \boldsymbol{a} & \text{at probability } \epsilon \end{cases}, \tag{2}$$

where $\mathcal{Q}$ is the network we learn in order to approximate the Q-function. Its implementation takes $\boldsymbol{s}$ as input and returns a vector of Q-values for each index $i$, indicating the Q-value obtained for each action $j = 0, 1, \ldots, i$.

$$\mathcal{Q}(\boldsymbol{s}, \boldsymbol{a}) := \sum_i \mathcal{Q}(\boldsymbol{s})[i, \boldsymbol{a}_i], \tag{3}$$

where indexing occurs first for the vector of Q-values per each index $i$ and then for an element in this vector. Therefore, the input and output domains of the approximated Q-function are $\mathcal{Q} : \mathbb{R}^{L-1} \to \mathbb{R}^{\frac{(L+2)(L-1)}{2}}$. This reflects the fact that $\boldsymbol{s}_0$ is fixed and that for every layer $i = 1, 2, \ldots, L - 1$ the network $\mathcal{Q}$ needs to assign values to $i + 1$ different actions. The optimal action-value function $Q^*$ obeys an important identity known as the Bellman equation

$$Q^*(s, a) = \mathbb{E}_{s'}[r + \gamma \max_{a'} Q^*(s', a') | s, a], \tag{4}$$

We run the policy $\pi$ based on $\mathcal{Q}$ to obtain a new action $\boldsymbol{a}$ after every $k$ training steps of the primary network $\mathcal{T}$. $k$ is relatively small and such actions are taken frequently. At initialization, only layer $i = 0$ is trained; all other layers are fixed at their initial values. Then, after $k$ training steps, and every $k$ training steps afterwards, we perform the following set of actions: (i) obtain a new action $\boldsymbol{a} = \pi(\boldsymbol{s})$, (ii) extract the new state $\boldsymbol{s}'$ based on $\boldsymbol{a}$, as in Eq. 1, (iii) replicate the weights of each layer $i$ to be the same as $\boldsymbol{s}_i$ and tie these weights, (iv) compute a reward for $\mathcal{T}$ based on the negative perplexity score as computed on a random training batch, (v) update $\mathcal{Q}$ based on the expected reward vs. the computed one, using the Bellman equation, (vi) reduce the exploration factor $\epsilon$ by a fixed factor of 0.95, but always keeping it above a constant of 0.1, and, finally, (vi) run $k$ more training steps for $\mathcal{T}$ and repeat.

The method is depicted in Alg. 1 and a line-by-line description is provided in Appendix A. A few implementation details are worth noting. First, in line 2, the replication pattern of the first $k$ steps (where $k << K$) is determined to be such that layer 0 trains and the other layers are kept fixed at their initialization values. Then, every $k$ steps we obtain a new action $\boldsymbol{a}$, using the $\epsilon-$greedy policy in Eq. 2, see lines 7-8. Second, we note that a layer that changes state can, based on the condition in line 12, either (i) shift from being untied or tied to one layer to being tied to a new layer, or (ii) shift to being trained independently. In the first type of shift, a new set of weights would be copied, which may change the transformer much more quickly than through gradient steps. In the second type of shift, the weights are not changed immediately. However, they begin to drift between layers that were previously tied. Third, the exact schedule for modifying the value of $\epsilon$ is given in line 27.

## 4 Experiments

In our experiments, two architectures were used: (i) GPT-2 with 48 decoder blocks, each with 16 attention heads. The hidden dimension for each block was set to 1600, and (ii) BERT, which consists

Table 1: Metric scores for the GPT-2 architecture

| Metric | Method | Training set | | | |
|---|---|---|---|---|---|
| | | Wiki-2 | Wiki-103 | Lambada | 1-billion |
| Perplexity | Conventional training | 53.57 | **22.32** | 94.96 | 88.35 |
| | Our method | **49.37** | 22.35 | **93.84** | **72.35** |
| Number of trainable parameters | Conventional training | 1.6B | 1.6B | 1.6B | 1.6B |
| | Our method mean over training | 171M | 151M | 166M | 218M |
| | Our method at end of training | 264M | 142M | 326M | 203M |
| Number of independent layers | Conventional training | 48 | 48 | 48 | 48 |
| | Our method mean over training | 4.395 | 2.309 | 3.547 | 4.486 |
| | Our method at end of training | 7 | 6 | 9 | 10 |
| Training time per epoch (seconds) | Conventional training | 148.5 | 3609.5 | 26376.5 | 15440 |
| | Our method | 165 | 4010.5 | 29307.2 | 17155.5 |

Table 2: Metric scores for the BERT architecture

| Metric | Method | Training set | | | |
|---|---|---|---|---|---|
| | | Wiki-2 | Wiki-103 | Lambada | 1-billion |
| Perplexity | Conventional training | 70.15 | 154.2 | 202.70 | >1000 |
| | Our method | **69.27** | **132.6** | **156.30** | **215.50** |
| Number of trainable parameters | Conventional training | 376M | 376M | 376M | 376M |
| | Our method mean over training | 52M | 52M | 52M | 57M |
| | Our method at end of training | 46M | 46M | 67M | 60M |
| Number of independent layers | Conventional training | 12 | 12 | 12 | 12 |
| | Our method mean over training | 2.36 | 2.83 | 1.88 | 2.45 |
| | Our method at end of training | 3 | 5 | 3 | 3 |
| Training time per epoch (seconds) | Conventional training | 51.2 | 1244.6 | 5324.1 | 9095.3 |
| | Our method | 26.5 | 644.5 | 2757.1 | 4704.4 |

of 12 decoder blocks with a hidden size of 768 and 12 attention heads at each layer. In all of our experiments, $\mathcal{Q}$ is an MLP with one hidden layer with 128 units and the ReLU activation function.

We ran all experiments for $K = 300$ epochs, a batch size of 16, and $k = 15$ with a separate validation set used to select the best model. The hyper-parameters used were: the transformer learning rate is set to 0.0001 and $\mathcal{Q}$'s learning rate was set to 0.001, $\gamma = 0.99$, the initial exploration probability is set to $\epsilon = 1.0$ (explore), and as depicted in Alg. 1, the $\epsilon$-decay factor: 0.95, and the minimal $\epsilon$ value is set to 0.1. Our experiments ran on 2-4 A100 GPUs for the GPT-2 based architecture and 1-4 A6000/A5000 GPUs for the BERT architecture.

**Datasets** In this study, we employ four widely used datasets to evaluate the performance of our method for language modeling tasks. All datasets were pre-processed by converting the text into tokens using GPT-2's tokenizer, which has a vocabulary of $50,257$ tokens. **WikiText-2 (Wiki2)** is a large language modeling corpus that consists of over 2 million tokens. It is derived from a snapshot of verified Good and Featured articles on Wikipedia. The dataset is widely used for training language models and serves as a standard benchmark for evaluating various NLP algorithms. **WikiText-103 (Wiki103)** is an extension of the WikiText-2 dataset, containing more than 100 million tokens. It is also sourced from Wikipedia articles and is considered to be one of the most comprehensive datasets for training large-scale language models. **LAMBADA** is designed to test the capabilities of language models in predicting the final word of a sentence, given all the preceding words in that sentence. The dataset contains approximately 10,000 examples, each a sequence of sentences extracted from books. The task is challenging as it often requires understanding the broader context provided by the preceding sentences. The **1 Billion Words** dataset is a corpus of text containing approximately 1 billion tokens, sourced from news articles. It provides a diverse range of vocabulary and sentence structures, making it ideal for training robust language models.

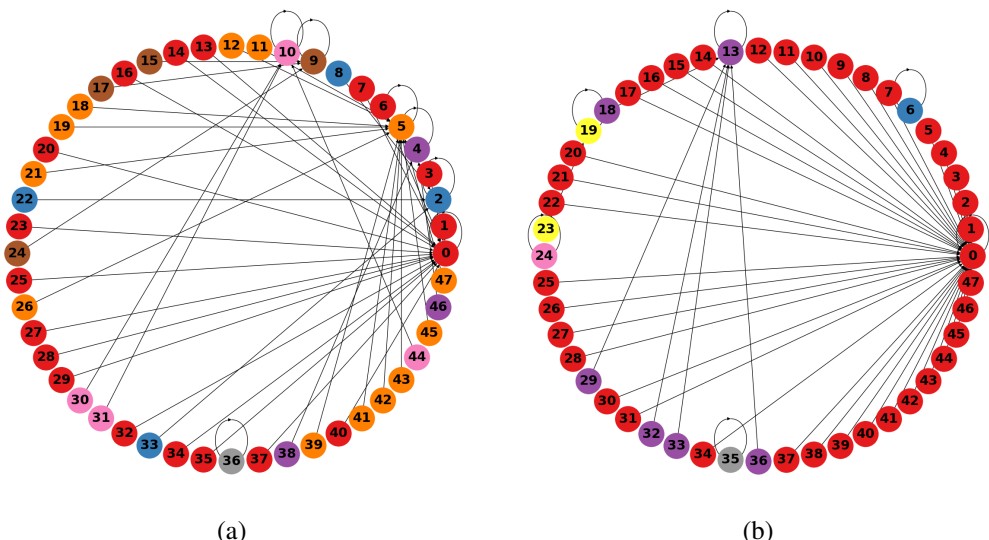

(a)  (b)

Figure 1: The replication map for the GPT-2 architecture post-training for (a) Wiki-2, (b) Wiki-103. The lowest-index layer in each group of layers that share weights is connected to itself.

**Results**    In Table 1, we present a comprehensive evaluation of our proposed method against conventional training on the GPT-2 architecture across multiple datasets: Wiki-2, Wiki-103, Lambada, and 1-billion. Our method consistently outperforms the baseline in terms of perplexity, with the most significant gains observed in the 1-billion words dataset, where we reduce the perplexity from 88.35 to 72.35. Additionally, our method exhibits a significant reduction in the number of trainable parameters, with a mean over training as low as 151M for Wiki-103, and not much higher on the other datasets, compared to the baseline's 1.6B. Although the conventional method outperformed our method on Wiki-103, the gap is marginal.

Table 2 showcases the results for the BERT architecture, presenting similar trends. Our method outperforms the conventional training across all datasets. Notably, in the 1-billion dataset, the perplexity is reduced drastically, from over 1000 in conventional training to 215.50 in our method. The number of trainable parameters also sees a substantial decrease, with a mean during training of 52M-57M, compared to the conventional 376M. In both architectures, we can observe that the mean number of independent layers (or, equivalently, the number of groups of identical layers) is rather low during training and is somewhat higher in the final model. Especially in BERT, we can observe that even for large datasets the number of independent layers is small. In our ablation study below we check whether one can simply train much less layers.

With respect to training time, the results are mixed. While in Tab. 2 it is demonstrated that our method somewhat slows down the training time, Tab. 1 presents a reduction of almost 50% in runtime. We believe, but have not yet verified, that this is due to the difference in hardware between the two experiments (GPT-2 runs on A100, the BERT runs on A6000/A5000).

The status at the end of the training is shown in Fig. 1. A line is drawn between every layer index $i$ and the layer it replicates $s_i$. A layer $i$ for which the state vector satisfies $s_i = i$ is connected to itself. As shown, there are seven such layers for Wiki-2 and six for Wiki-103, matching the statistics report in Tab. 2. The dominance of layer zero is clear, see Sec. 5 for a discussion of this property and its implications.

**Training dynamics**    The training process takes place under the guidance of a policy that is trained from scratch. This policy can change the layer topology drastically and it is, therefore, interesting to explore the training dynamics. First, it is not clear whether any changes are made at all to the topology throughout the training process. It could be the case that after a certain period of exploration, the policy is to keep the state fixed from one step to the next. As Fig. 2 demonstrates, this is not the case. We distinguish two types of state-change events, as detailed in Sec. 3. In the first, which we called "tied events", a layer $i$ replicates a layer it did not replicate previously. In the second type, termed "untied events", a layer $i$ obtains a new state of $s = i$ and is trained independently of

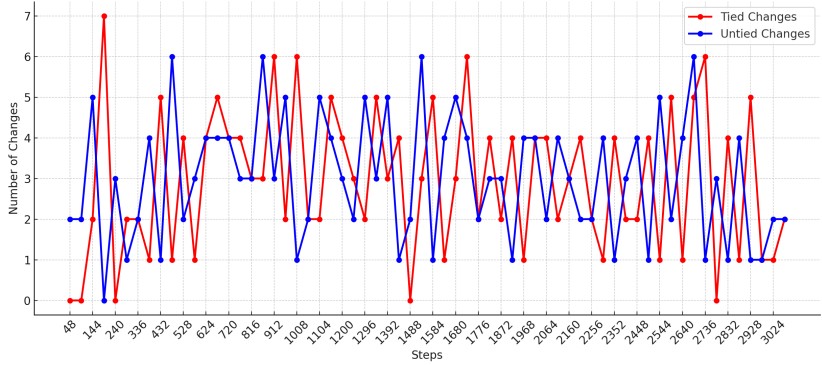

Figure 2: The number of change state events per type for training GPT-2 on Wiki-2

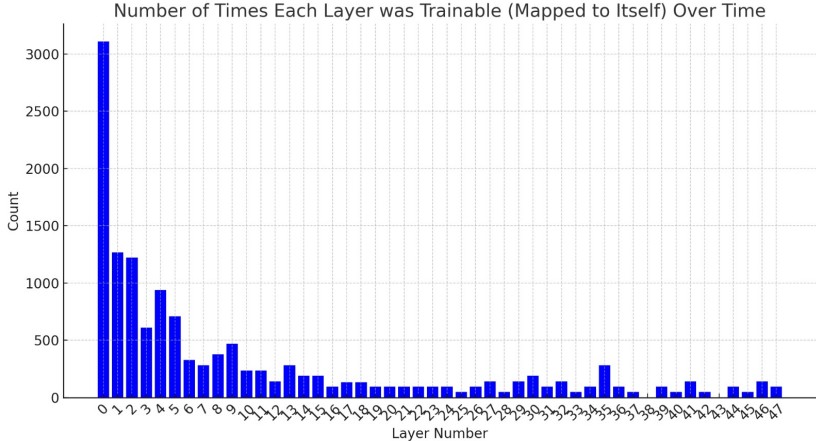

Figure 3: The number of steps in which each decoder block was trainable

Table 3: Memory consumption during training (GPT-2; batch size of 16; sequence length of 256).

| Statistics | Conventional training | Our method |
|---|---|---|
| Peak memory | 12,566.66 MB | 4,514.31 MB |
| Average memory consumption | 10,223.08 MB | 3,395.16 MB |

previous layers, which had replicated another layer $j < i$. Evidently, both types of events continue to occur throughout the training process and their frequency does not diminish.

The memory consumption during training is a result of the training dynamics. Tab. 3 depicts the peak and average memory consumption during the training of GPT-2. Our memory consumption is lower by 65% in peak consumption and 68% on average. This difference is obtained without any attempt to optimize memory usage during training or to release unused memory, and does not reflect in full the drop in the size of the model.

One may wonder if all layers have the same chance of being untied. We note that since the exploration factor $\epsilon$ is at least 0.1 throughout training, with the exception of layer 0, all layers are expected to be tied to other layers at one point or another. As can be seen in Fig. 3, this is indeed the case.

It can also be observed that the lower layers are more likely to have an untied status of $s_i == i$ (other layers with index $j > i$ may still have $s_j = i$ and train together in a tied way). This makes sense due to the increasing number of replication options that higher layers have. However, we note from Fig. 1(b) that the layers with $s_i == i$ can be relatively evenly distributed at the end of training.

**Ablation study**  Ablation experiments were conducted on the Wiki-2 dataset with the GPT-2 architecture. Since much of the ablations focus on validating that the success of the method does not arise from avoiding overfitting by reducing the network capacity, we also run ablations on the small Shakespeare dataset. This dataset has parts of Shakespeare's plays, sonnets, and other writings. It is small, with 250K tokens and the ablation uses a 12-layer GPT-2 like model.

Table 4: Perplexity scores for the ablation study for the Wiki-2 and the Shakespeare datasets.

| Architecture | Wiki-2 | Shakespeare |
|---|---|---|
| (i) Vanilla transformer $L$ = #independent layers in ours | 54.64 | 167.3 |
| (ii) Training all epochs with the final architecture | 59.35 | 185.3 |
| (iii) Applying the recorded dynamic on permuted indices | 65.21 | 172.2 |
| (iv) Applying the recorded dynamics on the indices without Q | 50.05 | 161.3 |
| (v) Fully dynamic without weight tying | 235.8 | 202.9 |
| (vi) All layers are trainable at initialization | 50.18 | **159.3** |
| (vii) "Cycle": Connecting layer $i$ to layer $\frac{L}{2} + i$ | 51.93 | 176.6 |
| (viii) "Cycle Rev": Connecting layer $i$ to layer $L - i$ | 52.83 | 180.0 |
| (ix) "Sequence": Connecting pairs of consecutive layers | 54.01 | 173.9 |
| Our full method | **49.37** | 161.1 |

Since the model obtained with our method has about a sixth of the number of parameters in the original model, we need to explore whether the full model capacity is required at all. To validate this, we designed a few ablations: (i) a transformer in which the number of layers $L$ is the number of independent layers obtained by our method, and (ii) training from scratch a static transformer architecture that has the same weight-tying structure as our method's final architecture.

As can be seen in Tab. 4, both these transformers are far behind our full method's results and also behind the conventional training results. The second ablation implies that our method is not suitable for finding "lottery tickets", i.e., pruned architectures for training from scratch (Frankle & Carbin, 2018; Chen et al., 2020a;b; Prasanna et al., 2020; Movva & Zhao, 2020).

Another related ablation (iii) checks whether the dynamic status changes can be made arbitrarily, by recording the state vector $s$ during the course of training, and applying a permuted version of it $\pi(s)$ when changing the status of a layer to copy another layer or to be tied, where $\pi$ is a fixed permutation operator that is applied element-wise.

The results of this ablation demonstrate that the layer identity is important and that a significant degradation occurs in the model's performance when the same dynamics are applied to a different set of layers. As a sanity check, we also (iv) run the recorded set of states on another training session (without performing Q-learning). As can be seen, this obtains results that are similar but slightly worse than those of the full unablated method.

The necessity of weight tying is demonstrated by ablation (v), in which weight replication occurs as in the full method, but weight tying does not take place. This leads to very unstable training and a very high perplexity score.

We also explore (vi) the effect of freezing all layers except for layer 0 at initialization by freeing all layers to train (removing line 2 of Alg. 1. This somewhat outperforms the full method on the shakespeare dataset but is less successful on Wiki-2. We conclude that freezing at initialization may not be crucial (more experiments are needed). However, it has a sizable advantage in the peak GPU memory consumption.

We also provide results for (vii) using half the layers and tying every layer $i = 1, 2, \ldots, L/2$ to layer $L/2 + i$. This cuts the number of trained layers by a much smaller fraction than our own method and is given as a reference since it was outlined as motivation in Sec. 1. As mentioned, this improves perplexity over the conventional training, but not nearly as much as our full method.

As mentioned in Sec. 2, ablation (vii) is the Cyclic pattern of (Takase & Kiyono, 2021). The two other patterns there are provided for completeness as ablations (viii) and (ix). As can be seen, these patterns, which reuse only 50% of the layers, are not as effective as our method.

## 5 DISCUSSION AND LIMITATION

Replacing the weights of an entire layer with those of another is a drastic change to the network. Yet, as shown in Fig. 2 (blue graph), such changes occur throughout training. This ability to perform this change without causing a temporal setback to the training process is not trivial, since even functionally equivalent layers can be expressed in multiple ways, by permuting the attention heads or the outputs of the feed-forward network. However, permutation to the feed-forward network would

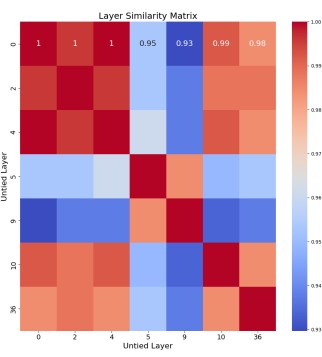
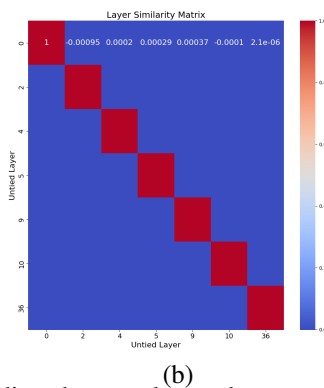

(a)                                 (b)

Figure 4: (a) Pearson correlations between the weights of the feed-forward networks of the untied layers (Wiki-2; GPT-2 architecture). The colorbar range is [0.93,1] (b) As a reference, the correlations between the same layers in the conventionally trained GPT-2 model. The value range is [0,1].

drastically modify the token embedding the next layer observes, and would cause the network's performance to degrade unless the other layers co-adapt.

We attribute the fact that no such setbacks occur to the way the training process initializes. Layer 0 trains in a way that cannot be too specific, due to the randomly initialized filters downstream, which require time to co-adapt. Then, layer 0 is replicated and multiple copies of it are trained simultaneously. Other layers are also copied and their copies begin to train. However, given that layer zero is a valid replication source for all layers, and given that the exploration constant $\epsilon$ is initialized at a high value, layer zero is dominant. This domination, as can be seen in Fig. 1, is maintained until the end of training.

We posit that all layers are exposed directly or through a replication chain to the information of layer 0, and that it spreads a specific order of attention heads and embeddings that are maintained across layers. Having this global alignment is crucial for smooth training despite large blocks of weights being copied during the process. Support for this hypothesis can be seen in Fig. 4(a), which depicts the Pearson correlations between the weights of the feed-forward networks of the independent transformer layers trained with our method. The minimal correlation is 0.93. For reference, the correlation between the same layers in the conventional training (some of the 48 untied layers) is shown in panel (b). The inter-layer correlations are close to zero, as expected by the arbitrary permutation argument.

Our research is focused on training transformer models from the ground up, contrasting with the extensive body of work that primarily concentrates on the fine-tuning of pre-trained transformers. (Devlin et al., 2018; Liu et al., 2019; Dodge et al., 2020; Raffel et al., 2020; Brown et al., 2020; He et al., 2021). It is unclear whether a method that starts with one trainable layer and then gradually explores options to untie some layers can be applied in such a case, especially since, as shown in Sec. 4, the number of independent layers remains small throughout training. An alternative that makes sense, but which is left for future work, is to apply the dynamic weight tying to the low-rank updates (LoRA) of Hu et al. (2021). One can also try to apply RL methods that employ backtracking (Dary et al., 2022), or use alternative search strategies, such as CAB (Zhang, 1998) or MCTS (Chaslot et al., 2008), changing one state index at a time.

The evaluation of our work is limited to transformers in the language domain. However, transformers are ubiquitous. A preliminary computer vision experiment reinforcing our conclusions can be found in Appendix B. Finally, transformers are often finetuned on downstream tasks. Preliminatry results on the GLUE set of benchmarks Wang et al. (2018) are presented in Appendix C, demonstrating that the tied models can be effectively trained for downstream tasks.

## 6 CONCLUSIONS

We present a method that is, as far as we can ascertain, the most dynamic form of Neural Architecture Search presented. During the training process itself, a deep Q-learning network drives a layer replication process, which ends up with over 90% of the parameters being in layers that completely replicate an earlier layer. This order of magnitude reduction in the number of parameters is achieved without sacrificing the perplexity score and, in some cases, also leads to an improvement in this metric. These surprising findings are further explored by visualizing the dynamics of the training process and the crucial components of the method are demonstrated in an ablation study.

## ACKNOWLEDGMENTS

This work was supported by a grant from the Tel Aviv University Center for AI and Data Science (TAD).

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

Table 5: The results of applying our method to ViT on CIFAR-10

| Metric | ViT | Our |
|---|---|---|
| Accuracy | 0.999 | 0.995 |
| # trainable params (mean) | 630M | 80M |
| # trainable params (end of training) | 630M | 139M |
| # trainable layers (mean) | 32 | 5.5 |
| # trainable layers (end of training) | 32 | 7 |

## A   A LINE-BY-LINE EXPLANATION OF THE METHOD.

The method is depicted in Alg. 1. In line 2, the replication pattern of the first $k$ steps is determined to be such that layer 0 trains and the other layers are kept fixed at their initialization values. The closest-matching action vector $a$ and stage vector $s$ are set to be the all-zero vector, see line 3.

The method then iterates over the data set and performs a regular training step on $\mathcal{T}$, see lines 5-6.

Every $k$ steps (where $k << K$) we obtain a new action $a$, using the $\epsilon-$greedy policy in Eq. 2, see lines 7-8. In line 9, we compute the state $s'$ based on the obtained action $a$ according to Eq. 1. This state is acted upon by replicating and tying the weights in lines 12,13. The condition in line 11 ensures that this happens only when this exact replication did not occur in the previous state $s$.

We note that a layer that changes state can, based on the condition in line 12, either (i) shift from being untied or tied to one layer to being tied to a new layer, or (ii) shift to being trained independently. In the first type of shift, a new set of weights would be copied, which may change the transformer much more quickly than through gradient steps. In the second type of shift, the weights are not changed immediately. However, they begin to drift between layers that were previously tied.

In lines 20-21 a random mini-batch is used to estimate the perplexity (PPL) score of $\mathcal{T}$. The reward $r$ is set in a similar fashion to the reward of DQN (Mnih et al. (2013)) as the sum of the evaluation score and the discounted prediction of the next state $\mathcal{Q}(s')$ (lines 22, 23). We do so by using Bellman (Eq. 4) as an iterative update: $\mathcal{Q}_{i+1}(s, a) = r + \gamma \max_{a'} Q_{i+1}(s', a')$.

The $MSE$ loss is then used to update network $\mathcal{Q}$ in lines 24, 25. As mentioned, after every training step of the Q-network, the value of $\epsilon$ is modified to balance exploration vs. exploitation, see line 27.

## B   PRELIMINARY COMPUTER VISION EXPERIMENTS.

Since transformers are ubiquitous, evaluating our method only for transformers in the language domain constitutes a limitation. As a preliminary computer vision experiment, we have applied our method to the Vision Transformer (ViT) (Dosovitskiy et al., 2021) on the CIFAR-10 dataset (Krizhevsky et al., 2009).

The results are reported in Table 5. As can be seen, similarly to the NLP experiments, with an insignificant drop in accuracy, our model has only 22% of the original model's parameters and only 7 out of 32 layers are independent at the end of training.

## C   PRELIMINARY DOWNSTREAM TASKS EXPERIMENTS.

In the domain of NLP, transformers are often trained for a causal language modeling task on a large corpus and are then fine-tuned on a smaller dataset for a specific task such as sentiment analysis, questions answering, or named entity recognition.

As a preliminary downstream task experiment, we have taken our GPT-2 based model which was trained using our method the 1-billion word dataset and trained it on multiple GLUE tasks Wang et al. (2018). During training on the new tasks, the tied layers were kept as in the final state of the model and the language modeling head was replaced with a new trainable head suited for each task.

Table 6: The results of finetuning the GPT-2 model trained on the 1-billion word dataset on multiple classification benchmarks. The number of trainable parameters and number of independent layers are at the end of training on the 1-billion word dataset and the subsequent finetuning, in which the tying of the layers is fixed.

| Metric | Conventional | Our |
|---|---|---|
| SST-2 (Accuracy) | 0.811 | 0.799 |
| Cola (Accuracy) | 0.691 | 0.691 |
| QNLI (Accuracy) | 0.608 | 0.599 |
| MRPC (Accuracy) | 0.697 | 0.697 |
| RTE (Accuracy) | 0.527 | 0.541 |
| # trainable params | 1.5B | 235M |
| # trainable layers | 48 | 5 |

As the vanilla baseline, we also trained a conventional GPT-2 model on the 1-billion word dataset and then finetuned all layers.

The results are reported in Table 6. As can be seen, our method leads to a minimal drop in the given metrics compared to the conventional method, while having only 12% of the trainable parameters and only 5 out of 48 layers are untied.

