# OpenReview forum: "Dynamic Layer Tying for Parameter-Efficient Transformers"
_ICLR.cc/2024/Conference — ICLR 2024 poster_

### Official Review · Reviewer_zEuz · 2023-10-28

**Soundness:** 2 fair
**Presentation:** 2 fair
**Contribution:** 2 fair
**Rating:** 5
**Confidence:** 4

**Summary:**

The paper proposes to share the weights across layers when training Transformer architecture from scratch. The proposed method utilizes a network Q, which is trained with Q-learning, to assign the sharing policy. Experiments on several text datasets show that the proposed method uses less trainable parameters, reaches comparable performance as the conventional training, and sometimes reaches better training speed.

**Strengths:**

The paper studies the weight-sharing scheme of training Transformers, and this may be used to reduce the training cost of developing large models.

**Weaknesses:**

There are two main weaknesses of the paper in my opinion.

1. The related work section is not complete and misses many relevant works and topics. I think several lines of research are very related to this paper but they are missing in the related work:

(1) network pruning (also lottery ticket hypothesis),

(2) weight sharing in Transformers,

(3) parameter-efficient fine-tuning methods (PEFT methods).

Network pruning is mentioned a bit in the introduction, but the paper doesn't mention the pros and cons of this paper against network pruning, making the contribution of this paper unclear. I also found that there are several shared-weight Transformers papers [1, 2, 3] that are not cited in the paper, and they should be discussed. The parameter-efficient methods (LoRA, Adapters, Prefix Tuning, Prompt-tuning, etc) save a significant amount of parameters (such as 1 - 5%) and are very relevant to this work.

[1] Subformer: Exploring Weight Sharing for Parameter Efficiency in Generative Transformers

[2] Sharing Attention Weights for Fast Transformer

[3] Lessons on Parameter Sharing across Layers in Transformers

2. Some weaknesses in experiments:

(1) Several weight-sharing techniques should be also included in the comparison.

(2) I would suggest reporting the performance on downstream tasks for a complete comparison. Lower perplexity sometimes does not mean higher downstream performance.

(3) The perplexity of the results seems very high compared to previous results. As far as I know, the perplexity of WikiText-2 is usually around 20 and the perplexity of 1-billion words is around 30. I would suggest finding the proper training recipe or training longer to let the model converge, or the current results may not be very convincing. The current better perplexity in Tables 1 and 2 might only be due to the faster convergence of the model with less trainable parameters.

**Questions:**

1. In Tables 1 and 2, why the training time of the proposed approach is a bit longer compared to the conventional training when training GPT-2 but is much less when training BERT?

---

> ### Author Response · Authors · 2023-11-21
>
> We thank the reviewer for the comprehensive and constructive feedback.
>
> The related work section has been significantly expanded following the review, and it now includes all the requested lines of work. We now explicitly differentiate our method from pruning methods, Reuse Transformer, and the methods and directions of research pointed to by the reviewer. Lottery ticket hypothesis was mentioned in our submission manuscript as part of the motivation in the introduction and now also in the text describing ablation (ii), which shows that our method does not lead to the lottery ticket property.
>
> > (1) Several weight-sharing techniques should be also included in the comparison.
>
>
> In Ablation (vii), we exactly replicate the Cyclic method from [1], which performs as well as the other two patterns explored in [1].
>
> The Reuse Transformer [2] only shares 10% of the weights, so is not a good baseline.
>
> Subformer [3] shares up to 50% of the weights (much less than what we share) and requires architectural changes that make it unsuitable for our experiments.
>
> Xiaoet al. [4] share attention weights and not layers (and also share much less than we do).
>
> We will add a modern pruning baseline to our experiments. However, the pruning typically reduces much less than what we share and is also a complementary technique to ours (the sharing and pruning can be applied side by side).
>
> [1] Takase, Sho, and Shun Kiyono. "Lessons on parameter sharing across layers in transformers." arXiv preprint arXiv:2104.06022 (2021).
>
> [2] Bhojanapalli, Srinadh, et al. "Leveraging redundancy in attention with reuse transformers." arXiv preprint arXiv:2110.06821 (2021).
>
> [3] Reid, Machel, Edison Marrese-Taylor, and Yutaka Matsuo. "Subformer: Exploring weight sharing for parameter efficiency in generative transformers." arXiv preprint arXiv:2101.00234 (2021).
>
> [4] Xiao, Tong, et al. "Sharing attention weights for fast transformer." arXiv preprint arXiv:1906.11024 (2019).
>
>
> > (2) I would suggest reporting the performance on downstream tasks for a complete comparison. Lower perplexity sometimes does not mean higher downstream performance.
>
>
> Due to constraints encountered during the discussion phase, we were unable to run additional experiments. We agree that it would be a useful experiment to conduct.
>
>
> > (3) The perplexity of the results seems very high compared to previous results. As far as I know, the perplexity of WikiText-2 is usually around 20 and the perplexity of 1-billion words is around 30. I would suggest finding the proper training recipe or training longer to let the model converge, or the current results may not be very convincing. The current better perplexity in Tables 1 and 2 might only be due to the faster convergence of the model with less trainable parameters.
>
>
> The perplexity results quoted are achieved by training the models on multiple datasets, not only on wiki/1b. For example, GPT-2 which was trained on multiple datasets without finetuning on wiki2 achieved a PPL of around 18 while the exact same model trained only on WIKI2 is around 50.
>
>
>
>
> > Question 1: In Tables 1 and 2, why the training time of the proposed approach is a bit longer compared to the conventional training when training GPT-2 but is much less when training BERT?
>
> This is addressed in the results section: “With respect to training time, the results are mixed. While in Tab. 1 it is demonstrated that our method somewhat slows down the training time, Tab. 2 presents a reduction of almost 50\% in runtime. We believe, but have not yet verified, that this is due to the difference in hardware between the two experiments. While GPT-2 experiments run on A100, the BERT experiments run on A6000/A5000.”

---

> ### Comment · Reviewer_zEuz · 2023-11-21
>
> Thanks to the authors for the response.
>
> * I checked the related work section. I found one typo "self-attention mechanism,m " (the m is extra) and also the parathesis for citation is missing.
>
> * Also, could you show me some references that also get around 50 perplexity on sole training on Wiki-2?
>
> * I am not satisfied with the explanation of the training time benchmark part, and the authors also don't know why this speed benchmark has mixed results. Now I am even more confused about whether the approach would make the training slower or faster because of this explanation, but it seems like we do know that the approach would make the training slower on either A100 or A6000.
>
> * Currently, I am still not totally convinced by the experimental results as several fixed pattern weight-sharing in [1] are not compared and there are no results on downstream tasks.
>
> [1] Takase, Sho, and Shun Kiyono. "Lessons on parameter sharing across layers in transformers." arXiv preprint arXiv:2104.06022 (2021).

---

> > ### Author Response · Authors · 2023-11-22
> >
> > We thank the reviewer for the constructive feedback and open discussion.
> >
> > > I checked the related work section. I found one typo "self-attention mechanism,m " (the m is extra) and also the parathesis for citation is missing.
> >
> > Thank you for checking the revised related work section. We apologize for these typos, which have been corrected in the revision.
> >
> > >Also, could you show me some references that also get around 50 perplexity on sole training on Wiki-2?
> >
> > We do not have a reference that trains GPT-2 directly on Wiki-2. The supplementary material now contains a notebook called gpt-2-on-wiki2.ipynb that recreates this experiment. It is based on [Kapathy’s code](https://github.com/karpathy/nanoGPT/blob/master/model.py).
> >
> > > I am not satisfied with the explanation of the training time benchmark part, and the authors also don't know why this speed benchmark has mixed results. Now I am even more confused about whether the approach would make the training slower or faster because of this explanation, but it seems like we do know that the approach would make the training slower on either A100 or A6000.
> >
> > You are right to be unsatisfied. As stated in the manuscript, the reasons for our method speeding up training by 50% on one architecture while slowing it down by 15% on another remain unclear. We need to further investigate it, but we have not been able to perform this task so far.
> >
> > However, please note that (1) we made no attempt to profile or optimize the code, (2) speeding up training is not one of our claims, (3) even in the worst case out of the two architectures, the change in training time is not very detrimental.
> >
> > > Currently, I am still not totally convinced by the experimental results as several fixed pattern weight-sharing in [1] are not compared and there are no results on downstream tasks.
> >
> > > [1] Takase, Sho, and Shun Kiyono. "Lessons on parameter sharing across layers in transformers." arXiv preprint arXiv:2104.06022 (2021).
> >
> > Following the review, we have added to the revision the two other patterns of [1] to the ablation table, alongside the one pattern that was already there. In the final version, we will run these on more datasets to add these baselines to the main tables. On the two datasets used for ablation studies, the two additional patterns have worse perplexity than our method and reuse only 50% of the layers.
> >
> > Due to challenging circumstances, we are currently unable to present results for downstream tasks. We acknowledge the extra validation that such results would offer.

---

> > > ### Comment · Reviewer_zEuz · 2023-11-23
> > >
> > > Thank the authors for the additional explanation and new experiment on sequential sharing architecture. I still have some concerns about the downstream evaluation and the mixed results on the training speed, but I will increase the score to 5 because the two fixed pattern baselines are compared and the related work section is expanded.

---

### Official Review · Reviewer_Mmpd · 2023-10-30

**Soundness:** 4 excellent
**Presentation:** 4 excellent
**Contribution:** 4 excellent
**Rating:** 10
**Confidence:** 4

**Summary:**

The authors propose a new means for essentially doing architecture search on a transformer model. The available variation for the model are which layers should be tied together, and which should remain independent.

They employ an RL algorithm to do the selection of what should be connected to what. The reward signal comes straight from the perplexity of a model after an action has been taken. The results are very exciting and significant. It is very possible that the authors have just opened the floodgates to a new research path on which many will walk in the near future.

The resulting models demonstrate that a transformer model with 48 layers, may require between 7-10 independent layer weights to perform as well, or better than conventionally trained 48 layer models. This opens up a lot of questions related to modern artificial neural circuitry and may allow for interesting modularization research to take place.

**Strengths:**

1. Clear, concise and precise writing.
2. The idea attempted itself is intriguing.
3. The idea is executed very well, resulting in excellent outcomes.
4. The results are very promising.
5. The results open up a lot of new questions, as well as motivate the need for research in the pathway the authors have opened up.
6. The paper represents what could be a seminal moment in architecture search methods, as well as understanding modularity in NNs.

**Weaknesses:**

1. The method section, while technically sound, suffers from a lack of clarity as to the method being presented. For such a significant contribution, it is important to ensure that one can understanding the fundamentals of the method without having to crunch through all of the equations presented and fill many of the gaps with their own imagination. Perhaps something like a functional diagram, a more intuitive algorithm, or just a page that lays out the ingredients one by one, as well as how they are optimized, followed by the algorithm page would serve clarity of communication better.
2. It's not clear if the authors have tried additional rewards in addition to PPL after updates, or if it was the only one tried.
3. Figure 1 is hard to read and understand. A more intuitive approach could leverage colour-coding to indicate a particular layers weights and perhaps a sequential figure showing a layer by layer colour coding might be more helpful/clear.

**Questions:**

1. Have you tried retraining the architecture that your method learned from scratch? As in, grabbing the connection maps, using that to initialize a new transformer and training from there. It would be interesting to how it compares with the model you trained jointly.
2. How did you try to balance exploration with exploitation? Presumably past actions can have a major effect on future trajectories. Did you employ any random restarts, or perhaps running multiple models in parallel so you can get feedback from a population rather than an individual for your RL updates.

---

> ### Author Response · Authors · 2023-11-21
>
> We thank the reviewer for the supportive review and constructive feedback.
>
>
> > 1. The method section, while technically sound, suffers from a lack of clarity as to the method being presented. For such a significant contribution, it is important to ensure that one can understand the fundamentals of the method without having to crunch through all of the equations presented and fill many of the gaps with their own imagination. Perhaps something like a functional diagram, a more intuitive algorithm, or just a page that lays out the ingredients one by one, as well as how they are optimized, followed by the algorithm page would serve clarity of communication better.
>
> We have considered this valuable suggestion. We agree that the writing style is layden with equations. Since another reviewer has asked to reduce the redundancy of the method section, we do not wish to add content there. Would it be acceptable if the text in the introduction serves as an overview? We can replace the 6th paragraph of the intro with a textual description, eliminating most of the equations there. We are also looking to create a video that demonstrates the stages of training and its evolution.
>
> > 2. It's not clear if the authors have tried additional rewards in addition to PPL after updates, or if it was the only one tried.
>
> We have not tried additional rewards in addition to PPL. This would indeed be interesting. Experience with RL in other domains suggests that combining rewards can be highly beneficial.
>
> > 3. Figure 1 is hard to read and understand. A more intuitive approach could leverage colour-coding to indicate a particular layers weights and perhaps a sequential figure showing a layer by layer colour coding might be more helpful/clear.
>
> Following the review, in the revised version we have added color coding to the existing graphs. Our initial attempts with a sequential ordering of nodes were not yet attractive enough to share, we will put additional effort into this.
>
> > Question 1. Have you tried retraining the architecture that your method learned from scratch? As in, grabbing the connection maps, using that to initialize a new transformer and training from there. It would be interesting to how it compares with the model you trained jointly.
>
> We believe that this is ablation (ii). This architecture when trained from scratch is less effective than a vanilla transformer with the same number of layers as the number of independent layers in it, as reported in ablation (i).
>
> > Question 2. How did you try to balance exploration with exploitation? Presumably past actions can have a major effect on future trajectories. Did you employ any random restarts, or perhaps running multiple models in parallel so you can get feedback from a population rather than an individual for your RL updates.
>
> All these are good suggestions. However, our training dynamics are exactly as given in Alg. 1. No random restarts or parallel runs were employed. We think these may be helpful for finetuning (we discuss finetuning in the last paragraph of Sec. 5).
> Exploration vs. exploitation was balanced using the epsilon greedy policy with a high epsilon at the beginning that favors exploration over exploitation. The schedule for updating $\epsilon$ is given in line 27 of Alg. 1.

---

### Official Review · Reviewer_xrb9 · 2023-10-31

**Soundness:** 3 good
**Presentation:** 1 poor
**Contribution:** 3 good
**Rating:** 6
**Confidence:** 3

**Summary:**

This paper proposes a method to reduce the number of trainable parameters of Transformers by dynamically tying the weights of different layers using reinforcement learning during the training process of Transformer. Experimental results indicate that compared to the conventional training method, the proposed approach effectively reduces the number of trainable parameters and slightly improves performance (perplexity score).

**Strengths:**

1 By employing a simple deep reinforcement learning algorithm, the number of trainable parameters of GPT-2 and BERT has been significantly reduced. The algorithm is easy to implement and holds the potential for application in other Transformer-based neural networks.

2 The algorithm has been thoroughly analyzed, providing a detailed explanation of the reasons for its effectiveness.

**Weaknesses:**

1 The Methods section contains a considerable amount of redundant content, providing a step-by-step explanation of the algorithmic process.

2 The experimental results only show the comparison of the proposed method and the conventional training method, lacking comparison with other baselines and the related methods.

3 The related work is not adequately summarized. There is only one publication after 2020 shown in the section of Related Work, and there is a lack of research focusing on improving the Transformer neural network architecture.

**Questions:**

1 The conclusion "Having this global alignment is crucial for smooth training despite large blocks of weights being copied during the process" seems to conflict with Table 3.(vi).

2 'While in Tab. 2 it is demonstrated that our method somewhat slows down the training time, Tab. 1 presents a reduction of almost 50% in runtime. We believe, but have not yet verified, that this is due to the difference in hardware between the two experiments. While GPT-2 experiments run on A100, the BERT experiments run on A6000/A5000.' Is there a citation error present?

3 The ultimate goal of reducing training parameters is to decrease memory? What is the actual consumption of memory?

---

> ### Author Response · Authors · 2023-11-21
>
> Thank you for the supportive review and constructive feedback.
>
>
> >1 The Methods section contains a considerable amount of redundant content, providing a step-by-step explanation of the algorithmic process.
>
> Thank you for this suggestion. The line-by-line explanation has been moved to Appendix A, and we kept in the main paper only the information that was not reported before this explanation.
>
> >2 The experimental results only show the comparison of the proposed method and the conventional training method, lacking comparison with other baselines and the related methods.
>
> We will add a modern pruning approach as an additional baseline to the final version. However, we note that such methods seldom prune more than 50% of the weights, which is much less than the reduction in the number of parameters we provide. Also, while pruning is a useful baseline for context, its contribution is orthogonal to ours and the two methods can be combined.
> We also note that ablation (vii) in our paper is similar to the cyclic sharing pattern of [1]
> [1] Takase, Sho, and Shun Kiyono. "Lessons on parameter sharing across layers in transformers." arXiv preprint arXiv:2104.06022 (2021).
>
>
> > 3 The related work is not adequately summarized. There is only one publication after 2020 shown in the section of Related Work, and there is a lack of research focusing on improving the Transformer neural network architecture.
>
> Thank you for your comment. Following the reviews we have added a considerable amount of related work to Sec. 2 of the manuscript. In the original submission, we have placed a lot of recent work in the first two paragraphs of the introduction, and we have now extended the related work section to better position and differentiate our work.
>
>
> > Question 1: The conclusion "Having this global alignment is crucial for smooth training despite large blocks of weights being copied during the process" seems to conflict with Table 3.(vi).
>
>
> With regards to this ablation study, we indeed write: ``We conclude that freezing at initialization may not be crucial (more experiments are needed). However, it has a sizable advantage in the peak GPU memory consumption.’’
>
> Following the review, we have revised the explanation in the discussion on how global alignment is obtained to discount the role of freezing. The revised hypothesis is that layer 0 cannot be too specific simply because there is no time for the other layers to co-adapt during the first k iterations.
>
> This is with regard to the role of freezing in obtaining alignment. The alignment itself exists, as we demonstrate in Fig. 4. The necessity of this alignment arises from the fact that without it, every time we would replace a layer (which can never stop occurring due to the exploration constant $\epsilon$), the performance would drastically drop (this theoretical argument appears in the first paragraph of the discussion).
>
>
> > Question 2: 'While in Tab. 2 it is demonstrated that our method somewhat slows down the training time, Tab. 1 presents a reduction of almost 50% in runtime. We believe but have not yet verified, that this is due to the difference in hardware between the two experiments. While GPT-2 experiments run on A100, the BERT experiments run on A6000/A5000.' Is there a citation error present?
>
>
> We are sorry, but we are not sure that we understand the comment regarding the citation error.
>
>
> > Question 3: The ultimate goal of reducing training parameters is to decrease memory? What is the actual consumption of memory?
>
>
> The model itself is reduced in size as reported in Tables 1 and 2, i.e., by up to 90%. Following the review, we report in the revised manuscript the memory consumption during training. In the current implementation, which does not actively release memory and was not optimized for memory consumption, the reduction is around 2/3 (Tab. 3).

---

### Official Review · Reviewer_ccf2 · 2023-10-31

**Soundness:** 3 good
**Presentation:** 3 good
**Contribution:** 2 fair
**Rating:** 6
**Confidence:** 3

**Summary:**

In this paper, the authors focus on the problem of reducing the number of parameters in transformers using Reinforcement Learning, specifically a variant of Q-Learning, to dynamically select layers in transformers and tie them together. They evalutate their approach on language models such as GPT-2 and BERT. They demonstrate the performance of their approach both in terms of final model performance and the amount of parameters reduced. They also conduct an ablation analysis of their approach.

**Strengths:**

1. Large transformer models have demonstrated excellent performance, but they are often very expensive in terms of computational resources. The paper addresses a very timely problem: ensuring transformer models can be applied in a practical setting without excessive cost.

2. The proposed method, parameter tying via Q-Learning-is intuitive and sensible approach for this problem.

3. Quantitative results both on the axis of performance and computational resources is convincing.

**Weaknesses:**

1. The approach is very specific in its focus. The architecture in question are only transformers, and evaluation is only on transformers in the language domain. It is unclear if these results hold for transformers in other data domains (such as vision transformers). It is also unclear if this approach (specifically parameter tying) could work for other neural architectures.

**Questions:**

1. The paper could be strengthened by evaluation on different data domains or perhaps even other neural architectures entirely. How would this approach perform on a vision transformer?

---

> ### Author Response · Authors · 2023-11-21
>
> Thank you for the supportive review and constructive feedback.
> Following the review, we ran ViT experiments on CIFAR-10. The results are reported in the table below and reinforce the manuscript’s results: with an insignificant drop in accuracy, our model has only 22% of the original model’s parameters and only 7 out of 32 layers are independent.
>
> | Metric                               | ViT   | Our   |
> |--------------------------------------|-------|-------|
> | Accuracy                             | 0.999 | 0.995 |
> | # trainable params (mean)            | 630M  | 80M   |
> | # trainable params (end of training) | 630M  | 139M  |
> | # trainable layers (mean)            | 32    | 5.5   |
> | # trainable layers (end of training) | 32    | 7     |
>
> We realize that CIFAR-10 is only a first computer vision experiment and therefore did not add it to the revised manuscript. Due to personal circumstances, we were not able to run as many experiments in the discussion period as we would normally do.

---

> > ### Comment · Reviewer_ccf2 · 2023-11-22
> >
> > Thank you for these additional experiments. Even though it's a simple vision dataset, I would suggest adding these results to the final manuscript if accepted.

---

> > > ### Author Response · Authors · 2023-11-22
> > >
> > > Thank you again for your valuable suggestion. We have uploaded a new revision in which we list the lack of computer vision experiments as a limitation and refer to the appendix, where the results of the ViT experiment are listed. We will make every effort to add more vision experiments to the final version.

---

### Meta-Review · Area_Chair_eLQu · 2023-12-09

**Metareview:**

The paper presents a method for dynamic weight sharing between the Transformer layers during training and uses reinforcement learning (Q learning) for realizing this. It shows the possibility of sharing a significant number of layers without much degradation in the final loss. Three out of the four reviewers are positive about the paper with one reviewer rating the paper as an important finding with the potential for opening up a line of follow up work. Reviewers had concerns on related work and limited scope of experiments to language, several of which the authors have answered with additional experiments and paper revision. I would strongly suggest the authors to follow up on their promised experiments and add these to the camera-ready version to address the remaining concerns from the reviewers.

**Justification For Why Not Higher Score:**

The paper still leaves some questions unanswered, eg, discrepancy in the training time for different model architectures as raised by Reviewer zEuz.

**Justification For Why Not Lower Score:**

Majority of the reviewers have rated the paper positively.

---

### Decision · Program_Chairs · 2024-01-16

Accept (poster)